# Guiding Post-Hospital Recovery by ‘What Matters:’ Implementation of Patient Priorities Identification in a VA Community Living Center

**DOI:** 10.3390/geriatrics8040074

**Published:** 2023-07-04

**Authors:** Katherine C. Ritchey, Laurence M. Solberg, Sandra Wolfe Citty, Lea Kiefer, Erica Martinez, Caroline Gray, Aanand D. Naik

**Affiliations:** 1Puget Sound Veterans Health Care System, Geriatric Research and Education Clinical Center (GRECC), Tacoma, WA 98498, USA; 2Division of Geriatrics and Gerontology, Department of Medicine, University of Washington, Seattle, WA 98109, USA; 3North Florida/South Georgia Veterans Health System, Geriatric Research and Education Clinical Center (GRECC), Gainesville, FL 32608, USA; 4College of Nursing, University of Florida, Gainesville, FL 32611, USA; 5Michael E. DeBakey Veterans Health Care System, Houston, TX 77030, USA; 6Palo Alto Veterans Health Care System, Palo Alto, CA 94304, USA; 7Institute on Aging, University of Texas Health Science Center at Houston, Houston, TX 77030, USA

**Keywords:** geriatrics, medical decision making, age-friendly health system, quality improvement, VA health system

## Abstract

Background: Patient priorities care (PPC) is an effective age-friendly health systems (AFHS) approach to aligning care with goals derived from ‘what matters’. The purpose of this quality improvement program was to evaluate the fidelity and feasibility of the health priorities identification (HPI) process in VA Community Living Centers (CLC). Methods: PPC experts worked with local CLC staff to guide the integration of HPI into the CLC and utilized a Plan–Do–Study–Act (PDSA) model for this quality improvement project. PPC experts reviewed health priorities identification (HPI) encounters and interdisciplinary team (IDT) meetings for fidelity to the HPI process of PPC. Qualitative interviews with local CLC staff determined the appropriateness of the health priorities identification process in the CLC. Results: Over 8 months, nine facilitators completed twenty HPI encounters. Development of a Patient Health Priorities note template, staff education and PPC facilitator training improved fidelity and documentation of HPI encounters in the electronic health record. Facilitator interviews suggested that PPC is appropriate in this setting, not burdensome to staff and fostered a person-centered approach to AFHS. Conclusions: The HPI process is an acceptable and feasible approach to ask the ‘what matters’ component of AFHS in a CLC setting.

## 1. Introduction

Older adults with three or more chronic disorders (i.e., multiple chronic conditions (MCC)) spend on average two hours a day on healthcare tasks, a half day per healthcare encounter and receive one guideline-recommended drug that harms a coexisting condition [1,2]. Though older adults with MCC vary in their healthcare preferences, most value maintaining function over a shorter life span and prefer therapies directed toward symptom relief over those which may extend life, especially if disease-modifying therapies might impact function or cause more symptoms [3]. Age-friendly health systems (AFHS) are intended to redesign healthcare delivery to address domains of health most important to older adults. These domains, known collectively as the ‘4Ms’ of geriatrics (Matters Most; Medications; Mentation; Mobility) encompass the medical, psychosocial and functional conditions which impact older adults in all healthcare settings [4]. Patient priorities care (PPC) is an effective approach to eliciting, documenting, and sharing patient priorities, which are derived from ‘what matters’ and align healthcare to meet the goals of older adults with MCC [5]. The PPC approach is an iterative process, which uses the context of the patient’s health trajectory (e.g., disease progression), prognosis, abilities, motivation and functional recovery (e.g., physical or occupational) to set specific, realistic and actionable goals honoring those values, consistent with their care preferences.

PPC is a structured process whereby a clinician or other trained facilitator completes health priorities identification (HPI) with a patient: a process that explores the patient’s values; translates values into specific, actionable, realistic health outcome goals; identifies the most bothersome symptom, healthcare task or life/social circumstance which is getting in the way of achieving that goal; elicits healthcare preferences (what one is willing and able to do or receive); and determines the ‘one thing’ (a symptom, health problem or burdensome task) that most interferes with the patient’s healthcare goal [6]. The information from the HPI process can then help clinicians consider if current care is consistent with a patient’s health priorities in the context of the patient’s health trajectory (e.g., disease progression) and then guide serial trials of starting, stopping or continuing medical interventions and help align care with health outcome goals which are intrinsically linked to ‘what matters’ (i.e., values) [7]. This requires consistent clinician–patient and clinician–clinician communication to find ways to best align healthcare [7]. The PPC approach is effective in achieving patient-centered care through the reduction in treatment burden and unwanted care (medications, tests, and self-care tasks) and increasing care that aligns with values in ambulatory care settings with older adults, including veterans [8].

The importance of priority-driven care is evident in subacute rehabilitation and skilled nursing facilities, such as VA community living centers (CLCs). In these settings, the prevalence of older adults with MCC and functional impairments requiring multiple therapeutic interventions or decisions is high [9]. Over the past two decades, VA CLCs have committed to providing high-quality, person-centered care in an effort to de-institutionalize nursing home care [10,11]. The PPC approach can reduce the treatment burden while aligning care to achieve healthcare goals that honor one’s values [8,12,13]. However, the feasibility of health priorities identification, let alone its influence on care alignment using the PPC approach in a skilled nursing facility (SNF) setting, has not been previously explored. Thus, the focus of this quality improvement project was to evaluate the fidelity, feasibility and acceptability of the health priorities identification (HPI) process from the PPC approach within the context of VA Community Living Centers (CLC). Through this work, our intention is to describe the logistical considerations, workflow, staff impressions and EHR tools necessary to systematically ask ‘what matters’ for veterans during their CLC admission and translate those values into specific, realistic and actionable health outcome goals.

## 2. Materials and Methods

### 2.1. Setting

The implementation of the HPI process was conducted as a quality improvement program in a single VA CLC located in Seattle, Washington. The CLC has 30 beds available for short-term rehabilitative or skilled nursing admissions. The unit is staffed with two physicians; two advanced practice nurses; physical, occupational, speech and recreational therapists; two social workers, a psychologist and a registered dietician. Nursing staff works as a team (one registered nurse (RN), one licensed practice nurse (LPN) and two to three nursing assistants (NAs)) overseeing twelve to fifteen patients. The average length of stay for veterans in 2021 was 32.3 days. Interdisciplinary team (IDT) meetings involved members of the care team from a variety of health professional disciplines (registered nurse and minimum data set nurse; occupational and physical therapists; physician; chaplain; advanced practice nurse and/or physician; psychologist; social worker) and the patient, themselves.

### 2.2. Health Priorities Identification (HPI)

The HPI process from PPC follows five steps where a facilitator (1) explores the patient’s values (i.e., what matters) within four domains shown to be most salient to older, medically complex individuals (connecting; enjoying life; managing health; and functioning) [3,14]; (2) translates values (i.e., what matters) into specific, realistic, and actionable health outcome goals; (3) identifies the most bothersome symptom, healthcare task or life/social circumstance which is getting in the way of achieving that goal; (4) elicits care preferences (i.e., how willing or unwilling a person may be to engage in a health behavior, treatment or therapy to achieve a particular health outcome goal [15]); and (5) determines the ‘one thing’ that a person wants to focus on so that they can reach their health outcome goal. The identification of the patient’s health priorities not only clarifies care preferences influencing patient and/or family decision making around healthcare, but documents how healthcare could be aligned to achieve the patient’s health outcome goals rooted in what matters [16]. The process could involve engagement with family or surrogate decision makers if/when cognitive impairment makes communication of health priorities difficult.

### 2.3. Health Priorities Identification (HPI) Implementation Strategy

#### 2.3.1. Implementation Team

A multidisciplinary team of national PPC and quality-improvement experts using a practice change framework guided the training and integration of the HPI process from the PPC approach into the CLC workflow [17]. The team consisted of physicians and advanced practiced nurses specialized in geriatrics, who have experience working in the CLC setting; are experts in PPC development and implementation; or are health systems researchers familiar with CLC workforce enhancement.

Individuals from this national expert team met with the CLC workforce and leadership to determine intervention readiness (i.e., willingness to adapt workflow and support staff to undergo training) of leadership, staff and setting. These pre-intervention meetings provided an opportunity for CLC leadership to learn about the PPC approach, outline current documentation processes for admissions and IDT meetings and determine how best to integrate HPI encounters into the normal CLC workflow.

#### 2.3.2. HPI Facilitator Training

The CLC leadership identified frontline staff who would be most appropriate to train as HPI facilitators, based on those persons’ time, willingness to participate and lead a quality improvement project, and experience and clinical background (e.g., motivated to expand communication skills and training; prior clinical training or experience in geriatrics or palliative care; well respected by their colleagues). Facilitators conducted HPI encounters for patients newly admitted to the CLC and represented a variety of health professions (registered nurse; occupational and physical therapist; physician; chaplain; advanced practice nurse; psychologist). The process for training has been described previously [13]. In short, facilitators reviewed asynchronous training videos (ranging from 20 min to 50 min), which provide a framework for eliciting patient health priorities using a motivational interviewing approach. The first video provides a PPC overview. The subsequent videos build the skills necessary for identification of health priorities: (1) explore values; (2) translate values into specific, realistic, actionable health outcome goals; (3) identify the most bothersome problem or symptom; (4) elicit care preferences (e.g., willingness or ability to engage in health behaviors); and (5) determine the ‘one thing’ that is getting in the way of the health outcome goal. Facilitators were also trained in strategies to align healthcare to achieve patient-directed goals, mostly provided as recommendations to the healthcare team. The facilitators encouraged IDT members to discuss considerations for healthcare alignment with the patient during IDT meetings. After completing the asynchronous training, the team of national experts in PPC provided in-person observations, EHR review and feedback to CLC-based HPI facilitators to enhance their skills and practice of the HPI process. Facilitators were independent in the HPI process after completing training videos and conducting two to three observed HPI encounters with feedback.

### 2.4. Implementation Measurement and Analysis

The focus of this quality improvement project was to evaluate the fidelity and feasibility of the health priorities identification process in a CLC setting. Structured meeting notes documented the progression through the first steps of each PDSA cycle (i.e., Plan; Do). The implementation of HPI into normal CLC processes was measured at the end of the first and second PDSA cycle (i.e., Study). Fidelity was assessed as the completeness of HPI documentation for each of the core steps mentioned previously (patient values; specific, actionable, realistic health outcomes; bothersome symptoms; healthcare preferences; the ‘one thing’) as well as integration of the health outcome goal and ‘one thing’ from the HPI into IDT meeting documentation. An EHR review was conducted by national experts in the PPC approach who evaluated all HPI discussions for encounter completion, quality of documented health outcome goal and ‘one thing’ and if the health outcome goal and ‘one thing’ was consistent with stated values, similar to prior implementation studies. This was performed using a validated chart review tool and PPC facilitators were aware of this process [18].

Intervention adoption (i.e., feasibility) was measured through the ability to complete HPI discussions with all patients newly admitted into the CLC and acceptance of HPI process integration into clinical assessments and workflow by staff. Qualitative interviews were conducted one-on-one with CLC-based HPI facilitators. All trained providers were invited to participate, but only five agreed to an interview. The primary interviewer was a medical sociologist with expertise in qualitative methods. Additionally, the interviewer was not part of the national expert implementation team and did not provide training and feedback. The interviewer used a semi-structured interview guide meant to elicit open-ended discussion of overall experiences with and reaction to the PPC intervention. Themes explored in these interviews included factors concerning the impact of PPC on workflow; barriers to implementation (staff buy-in, organizational barriers, patient-level barriers, others); HPI training and resources; recommendations for implementation and sustainment of the HPI approach in the CLC. Qualitative data were analyzed using an inductive rapid analytic technique and individual interviews were summarized in a spreadsheet according to the main thematic categories [19]. The study team then looked across interviews to identify patterns and recurring experiences and observations expressed across interviews. These patterns were then transformed into themes or a pattern iteratively identified in the data as an important research finding.

## 3. Results

Over the eight months of HPI process implementation, nine facilitators were trained and documented twenty HPI discussions with patients. HPI discussions occurred between patients and facilitators only and outside the context of the admission. The average length of stay for these patients was 56 days with one readmission to the hospital and no deaths. All facilitators completed asynchronous training videos, observation and feedback sessions during the first PDSA cycle. During the quality improvement program, there was no turnover of trained facilitators or IDT team members and only minimal turnover of CLC nursing staff not involved in the implementation project.

The first PDSA cycle included the first 10 HPI encounters, took an average of 40 min and occurred within the first week after admission. Review of the EHR indicated that the HPI process occurred in 60% of new admissions (Table 1). For patients who had an identification conversation, 66% of encounters included all core steps of the HPI process and 66% clearly documented the ‘one thing’ in the facilitator note. Twenty-two percent of facilitator notes had healthcare goals, which were specific, actionable, measurable and realistic. All were connected to the patient’s values. As for IDT care plan notes, 44% had documentation of health outcome goals that were consistent with the ‘one thing’ listed in the HPI facilitator notes.

The second PDSA cycle included developments to the program such as the development of a ‘Patient Health Priorities’-specific EHR template and notes (similar to: https://patientprioritiescare.org/wp-content/uploads/2021/12/Health-Priorities-Identificatin-template-note_Version2.pdf, accessed on 31 May 2023) (Figure 1); assigning facilitators to complete HPI encounters with each new admission; educational sessions for IDT members regarding the purpose of the program, overview of the HPI process within the PPC approach, location of HPI documentation and how to consider patient priorities during IDT care planning meetings (Table 1). With these changes, HPI encounters occurred with all patients within the first week of admission to the CLC (100%), but still took an average of 40 min to complete. The ‘Patient Health Priorities’ template and notes made HPI encounters easily identifiable in the EHR by all IDT members and facilitator documentation of the HPI core steps were complete for all encounters (100%). The quality of HPI documentation also improved. The quality of health outcome goals increased (66%) as well as the articulation of the ‘one thing’ (90%). Review of IDT care plan notes indicated an improvement in the consistency between the health outcome goals documented in the IDT notes and HPI encounters (78%).

Qualitative interviews with five facilitators indicated that the HPI process of PPC is very appropriate for the CLC setting, benefits patient care, and requires intentional leadership and interdisciplinary collaboration to achieve patient care alignment and build a more person-centered culture. Those interviewed felt that the HPI process of the PPC approach supported provider work (regardless of license or role) and at times made it easier (Table 2). The HPI process is sustainable and impactful as long as information gained from these discussions is incorporated in team-based meetings. Documentation is not burdensome but should be located in a consistent place in the EHR so that all providers can easily reference patient priorities. Leadership support and sponsorship facilitates program adoption and is essential for initial start-up and sustainability. Threats and barriers identified by those interviewed include staffing vacancies (causing cross-cover of duties and reduction in admissions), patient-related cognitive impairment (e.g., dementia, poor health literacy, psychological distress or serious mental health diagnosis), resistance to culture change (shift in way clinicians view and speak to veterans), and confidence in ability to probe if/when the conversation stalls (flexibility and creativity with exploration of values and priorities) (Table 2).

## 4. Discussion

We demonstrated that the HPI process of the PPC approach is a feasible and acceptable method for eliciting the ‘what matters’ component of AFHS in the CLC setting. Fidelity evaluation of HPI documentation also suggests that many different healthcare professionals can be trained in this process and determine health outcome goals and the ‘one thing’ rooted in what matters to facilitate person-centered care alignment in a CLC setting. The HPI process and documentation was not burdensome to staff, complemented normal workflow processes and was quickly mastered. Based on qualitative interviews, facilitators felt that the HPI process aligned with their clinical work, did not add significant time and provided a supportive framework for exploring what matters. The development of specific, measurable, realistic healthcare goals connected to stated values was improved by deliberate modifications in the second PDSA cycle (e.g., EHR tools and enhanced IDT member awareness of HPI encounters). Lastly, qualitative feedback from our program evaluation indicated that the HPI process changed how providers ‘see their patients.’ Facilitators shared that the HPI process of PPC helped them connect with a patients’ humanity and not see them as a combination of their health complications or functional limitations. Prior work in PPC and other serious illness communication programs supports this finding [12,20]. Setting aside time and space for conversations, which explore who a person is and what is most important to them, can develop a culture of person-centered care.

Patients in SNFs have medical and psychological complexity greater than community counterparts, have undergone a recent change in functional status and have a high risk of mortality [9,21]. Thus, it is appropriate for clinicians in these settings to reexamine if patient health priorities and their medical plan align if/when current care plans are discordant with a patient’s stated goals [22]. In 2015, the Agency for Healthcare Research and Quality (AHRQ) presented the SHARE approach, a streamlined shared decision-making model to help patients with their clinicians choose between evidence-based options, which are often in equipoise [23]. The SHARE approach comprises five steps (Seek, Help, Assess, Reach, Evaluate) with training that informs clinicians on how to perform each step [24]. PPC and SHARE approaches elevate the importance of providing care based on a person’s values and preferences [25]. However, the PPC approach provides the opportunity for healthcare providers to explore a person’s values and preferences in the context of their disease trajectory and clarify health outcome goals based on what healthcare tasks or treatments patients are willing or unwilling to participate in. Using the PPC approach to align care with ‘what matters’ can guide Age-Friendly Care delivered in any setting. 

Implementation of PPC into a setting such as an SNF requires commitment from leadership and staff to make integration of this approach feasible and sustainable. Lessons learned from our implementation evaluation mirror those observed in other serious illness communication programs and are helpful for a variety of healthcare dissemination practices and could guide PPC implementation in other non-VA settings [20]. We found that transparent and consistent EHR documentation elevated the significance of the HPI encounter and a patient’s stated values and goals. The use of EHR templates, trainings and feedback sessions strengthen facilitator skills in defining healthcare goals, which translated to clearer communication to IDT members. Implementation champions, or in this case, facilitators, should include persons who are motivated, respected by their colleagues, diverse in clinical and cultural backgrounds and willing to share the ‘work’ of implementation. Lastly, HPI encounters and documentation take time and may conflict with other work-related tasks and patient care. Leadership support is instrumental to overcoming the effort required for program integration that in the short-term competes with other setting-specific priorities, such as staffing, documentation, and other quality improvement and assurance initiatives. If the leadership team sets a clear tone of support and expectation of documenting change, then the implementation is more likely. Resources and time need to be allocated for staff training as well as encouragement for modifications to workflow or clinical tasks. Having multiple HPI facilitators of different clinical backgrounds was essential to the sustainability of this program in a CLC setting. It provided flexibility in times of staffing shortage, increased turnover and a diversity of clinical perspectives. Though the HPI process was not perceived as burdensome, sharing the role of HPI facilitation with others made integration more sustainable for the CLC staff. To help integration, the national PPC team has active websites with toolkits to support implementation in VA and non-VA settings (Implementation Toolkit—Patient Priorities Care; Patient Priorities Care VA Implementation Toolkit—Patient Priorities Care VA Implementation Toolkit).

There are several limitations to our study. Future studies will need to explore how HPI encounters and documentation can inform care alignment in the CLC and thus improve the patient experience of care vs. measures of quality of care. Though we reviewed integration of the HPI process of the PPC approach in IDT documentation, we did not analyze how the articulation of patient-stated priorities and goals was translated into distinct treatment decisions nor how those goals might need to change due to acute changes to health status (i.e., hospitalizations). A larger, prospective study with a control group could better evaluate if the PPC approach is associated with treatment differences observed in the medical record, such as length of stay, which was significantly longer during our QI intervention. Other systematic changes (e.g., changes to IDT documentation better incorporating patient preferences and priorities; staff education of the PPC approach; culture change and increased familiarity with HPI documentation) could improve the care alignment in IDT based on the HPI encounter. Second, our study was limited to a small VA-based CLC. Though there are many similarities between CLC populations and settings and community-based SNFs, it may be difficult to generalize our successes with implementation in other SNF settings where staff turnover might be higher. Lastly, our qualitative interviews were limited in scope and omitted the voice of the veteran and/or family. The interviewer did not oversee PPC training or feedback and interviews were blinded to the implementation team and CLC leadership. However, the low number of interviewees may bias the interpretations given and limit the generalizability of staff perceptions of the acceptability of this approach. Future work can explore the impact of the HPI process on patient/family perceptions of quality of care and other measures of ‘value-based’ care which impact SNF ratings, reimbursement and incentive payments.

## 5. Conclusions

Identification of health priorities using the core steps from the PPC approach is feasible in the context of normal CLC operations and can reliably translate ‘what matters,’ into specific, realistic and actionable health outcome goals. Implementation requires motivated workforce champions, supportive leadership and uncomplicated EHR tools. Our work will help guide other institutions interested in becoming Age-Friendly and incorporating PPC as their method for guiding patient care around ‘what matters’.

## Figures and Tables

**Figure 1 geriatrics-08-00074-f001:**
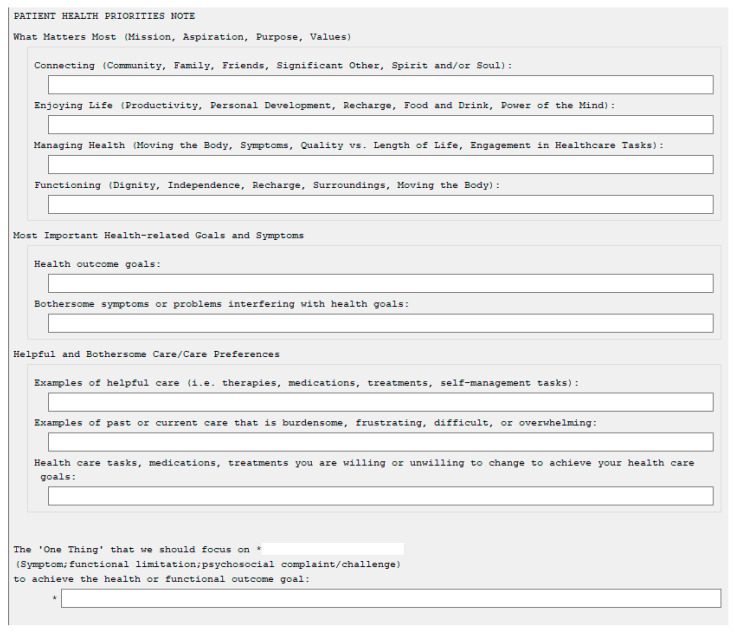
Patient Health Priorities Note Template Example with HPI Core Steps.

**Table 1 geriatrics-08-00074-t001:** PDSA breakdown and lessons learned from implementation evaluation.

Cycle 1	Cycle 2
Plan:	Plan:
Interface with local CLC staff and review of workflow Assess areas for improvement in patient engagement and care plan alignment Understand current challenges in documentation, team communication and care plan management Selection of front-line staff to champion program implementation Initial orientation and training of PPC model and process for eliciting patient priorities Create local CLC workgroups and shared working spaces for PPC training resources, templates and encounter tracking Engage leadership in program purpose and intervention plan	Meet with site implementation teams and draft changes to workflow Develop tools to track new admissions and assign PPC facilitation Engage with local EHR representatives for template development Formulate plan for engaging local CLC IDT team members on intervention purpose
Do:	Do:
Establish and train facilitators in PPC process Incorporate feedback to newly trained facilitators to ensure fidelity to the program Develop working flow for PPC conversations and integration into normal admission and IDT meetings Complete PPC conversations for persons who are newly admitted to CLC	Present to local IDT members intervention processes, documentation and purpose Encourage IDT members to review information elicited from PPC conversations Create PPC note template Train additional facilitators
Study:	Study:
Review PPC encounters for completion and content Review IDT documentation Regular local meetings to check in with workflow processes and PPC process integration Qualitative interviews with frontline staff	Review PPC encounters for completion and content Review IDT documentation Regular local meetings to check in with workflow processes and PPC process integration
Act/Lessons learned:	Act/Lessons learned:
Need a process for assigning PPC facilitators upon new admissions Expand numbers of PPC-trained facilitators Design and integrate PPC-specific EHR documentation and templates Leadership re-engagement with accomplishments	Necessary to have several PPC-trained facilitators Delegation of PPC conversation shares workload and clarifies responsibility Creation of EHR tools to support IDT member awareness of patient priorities Repeated engagement with staff about intervention purpose assists with model implementation and system change adoption by all team members

**Table 2 geriatrics-08-00074-t002:** Staff Impressions of HPI Integration into the Community Living Center Workflow.

Theme	Quote
Appropriateness for CLC	“PPC is good fit for CLC. When working with Veterans who are very depressed with psychiatric issues, it’s a way to remind them of what they talked about and what their goals are. She has noticed one veteran in particular—goal hasn’t changed, but every day is different. His goal is he wants to go home, but he has to get stronger and eat. It makes it easier having that conversation.” “Definitely thinks it’s a good fit. Thinks figuring out what matters most to patients can help with the connection between the provider and the veteran. It gives them something to come back to when talking about with Veterans about their care, especially with rehab.”
Helped or hindered ability to care for Veterans	“Learning what matters most helps connection between veteran and provider, and learning those important things about veterans helps build rapport. The veteran is more likely to remember who she is; it definitely helps with veteran engagement in their rehab.” “[PPC] It’s a great model. Sees what patients want for their healthcare gets overlooked. It can also help guide a general approach so are not inundating people with specialists and medications that aren’t necessary. “I’m excited about this kind of approach to help people take charge of their own healthcare.” “PPC has been helpful. Partly just because it helps us to identify things that are really important to them so we can help tailor our approach. I don’t think it’s hindered our ability to provide clinical care.” ““if anything it helps move the care forward and that’s why it’s so important.” Taking a wholistic approach to the patient—they are more than their amputation or the sprained limb. You have to take into consideration what’s important. Definitely a value added to their care.” “[PPC] has definitely helped. When you have these interdisciplinary teams working together, it helps make sure needs are met. Helps to make care more cohesive and flowing rather than staggered.”
Impact of HPI on workflow	“Took a little time to get acclimated and integrate PPC into normal routine. Estimated took about 30–45 min extra, but less now. The creation of a template with the [PPC] questions cut down a lot of time. Before I was typing in every question. Now can just put in the answers the veteran gives.” “I didn’t think it had impacted [my] work flow.” “For most part it aligns with what were already trying to do but it gives more of a framework. We focus a lot on geriatric care…. It gives us a good framework to have these conversations and help direct care.” “Didn’t think impacts flow—“it’s just the way I talk to them.” Doesn’t feel that different from what [I was] doing before, just worded a little bit different.” “Would think [it] wouldn’t take as long once [one] gets more used to the conversations because right now it is new. I will stumble less.”
Staff/System Barriers	“Trying to delegate the [HPI conversation] responsibility to one or two individuals identified the need for more HPI facilitators. It was important to workflow to acknowledge the time it takes to have those conversations and the time it takes to document those conversations.” “A lot of PPC is changing the way providers see things. Of course, as providers they have an idea of what is best for the veteran, but they have to remember to take in what’s important to the veteran as well. They can work together. It’s a shift in how they speak to the residents.” “Sometimes there was a time [barrier] if short staffed or more patients than normal, would probably leave the additional part out and make a note to come back. But this would be very rare.” “There are few barriers to implementing PPC. It’s just a different way to have a conversation. The end goal is getting the veteran to their better state of health whatever that looks like. Instead of telling them what to do because that’s what providers want.”
Patient Barriers	“PPC conversations can be challenging when veterans are very depressed. Getting through the conversation in one session is hard. Have to go back and try to reengage. Veterans weren’t able to sit with her and have the entire conversation.” “Patients have no problems answering the questions, unless they have dementia.” “Some patients have been challenging. One patient severely depressed and has cognitive issues, and it’s really hard for him to think about what would be his priorities outside of ‘I want to rest and lie in bed.’ Having those conversations can be extremely time consuming and difficult.”

## Data Availability

The data presented in this study are available on request from the corresponding author. The data are not publicly available due to the sensitive nature of staff interviews.

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
