# Peer review of "Guiding Post-Hospital Recovery by ‘What Matters:’ Implementation of Patient Priorities Identification in a VA Community Living Center"

_geriatrics, 2023, doi:10.3390/geriatrics8040074_

Round 1

Reviewer 1 Report

Page 3, line 104:  Suggest removing the comma between “multidisciplinary” and “team”

Page 3, lines 104-108:  This sentence is awkward to read.  Suggest changing “and had experience” to “who had experience, and adding the word “and” after the last semicolon in this sentence, and before “health systems researchers familiar with CLC workforce enhancement.” 

Page 3, line 116:  Suggesting changing “whom” to “who”

Page 3, line 124:  Suggesting changing “provides PPC overview” to ‘provides a PPC overview”

Page 3, lines 129 – 131: Please comment on potential limitation of conducting the process of aligning healthcare to achieve patient goals, as unlike PPC conducted in the outpatient setting, this alignment did not seem to involve the patient when done in the CLC setting.  While the structure of a patient visit in the CLC setting certainly differs from the outpatient setting, I am concerned that patients were not generally involved in this alignment step.

Page 3, lines 133 – 135:  The authors state that 2 to 3 HPI encounters were observed with feedback.  Please provide additional details about how this feedback was provided.  Were the PPC experts there via Zoom or at the bedside with the facilitators conducting the HPI, or was this observation conducted some other way?

Page 4, lines 128 and 144:  Suggest removing the hyphen between “one” and “thing” here and elsewhere in the manuscript.

Page 4, line 161:  The length of stay of the patients enrolled in this QI intervention (56 days) was much longer than the average length of stay in the CLC facility in 2021 (32 days).  Please comment on whether there were factors that may have led to this longer length of stay.

Page 4, line 164:  This CLC may be atypical in that there was minimal staff turnover.  Please comment in the Discussion section on how PPC can be sustained in the skilled nursing facility environment when there is greater turnover of certified nursing assistants.

Page 4, lines 165 – 166:  Please clarify whether the 40 minute time period was for the HPI component of the admission note, or if the 40 minutes comprised the time required to conduct both the admission H&P and the HPI.  [I believe the former is the case because on page 5, lines 179-180, it is stated that the 2nd PDSA cycle involved conducting HPI with each new admission.]

Page 5, line 183:  The authors state that in PDSA cycle 2, HPI was implemented into all new admissions.  I would recommend that the authors comment on how this was done and that they comment on the clinicians’ experience with building this into their workflow (including whether the clinicians perceived that using this approach added extra time to their day).  Please also indicate who was performing the new admissions – were these done by physicians only, or perhaps by both physicians and advanced practice providers?

Page 5, table 1, right-hand column (“Do”), 2nd bullet:  There is an extra space between “to” and “review” that should be removed.

Page 6, lines 192-208 (and Table 2, “Impact of HPI on Workflow”; also first comment in “Staff and System Barriers” and last comment in “Patient Barriers”):  Please expand on the comment that facilitators felt that documentation was not burdensome, as I think this is an extremely important factor in the ability to sustain PPC in this care setting. Did the qualitative interviews reveal anything about the actual time required to conduct the HPI and if the clinicians felt that HPI took too much time?

Page 7, Table 2, first row (“Appropriateness for CLC”):  There is an extra space between “goal” and “hasn’t”

Page 8, top row of the table:   In the passage reading “f short staffed,” please fill in the word starting with the letter “f.”

Page 8, “Patient Barriers” section:  One of the comments indicated that dementia was a barrier to the HPI approach.  Please comment on whether  family members and surrogate decision makers of CLC patients could be engaged to have the HPI discussion.

Page 8, lines 211-212:  Consider adding “in the CLC setting” at the end of this sentence “…for the feasibility of eliciting the “what matters component of AFHS in the CLC setting.”  Otherwise, I think this statement makes too broad of a claim in relation to the study design and setting.

Page 8, lines 224-225:  Suggest removing the comma after “conversations,” as this comma makes the sentence read somewhat awkwardly.

Page 8, line 234:  Consider changing the phrase “which are often equipoise” to “which are often in equipoise”

Page 9, line 251:  Suggest removing the comma between “and” and “documentation”.  Also, change “takes time” to “take time.”

Page 9, line 258:  Change “perspective” to “perspectives”

-Page 9, lines 259-279:  I would recommend adding the following points to the discussion section:

-A high “activation energy” was used to launch PPC at this CLC site, with a very intensive intervention including coaching form PPC experts.  This approach might not be realistic to apply more broadly.  Could the authors comment on how they can scale PPC in CLC and non-VA skilled nursing facility settings in a less intensive fashion (i.e. without the direct involvement of PPC experts?)

-I recommend commenting on implications of the HPI approach for billing in the long-term care setting.

Reviewer 2 Report

The study aimed to evaluate the feasibility and effectiveness of the Health Priorities Identification (HPI) process in VA Community Living Centers (CLC) as part of the Patient Priorities Care (PPC) approach to Age-Friendly Health Systems (AFHS). PPC experts collaborated with local CLC staff to integrate HPI into the CLC and utilized a quality improvement model. The study found that development of a Patient Health Priorities note template, staff education, and PPC facilitator training improved fidelity and documentation of HPI encounters in the electronic health record. The facilitator interviews showed that PPC is appropriate in this setting and fosters a person-centered approach to AFHS. The study concludes that successful integration of the HPI into a CLC and demonstration of using this approach to ask the ‘what matters’ component of AFHS is feasible.

1. What is the goal of the HPI process and how does it clarify care preferences?

2. Who led the training and integration of the HPI process into the CLC workflow?

3. Who determined the intervention readiness of staff and setting?

4.Who were trained as HPI facilitators and how were they selected?

5. What was the process for training HPI facilitators?

6. Who provided observation, EHR review and feedback to CLC-based HPI facilitators?

7. What was the focus of the quality improvement project?

8. What is the HPI process of the PPC approach, and how does it relate to person-centered care in SNF settings? 

9. What are some of the benefits and challenges of implementing this approach, and how can these be addressed? 

Reviewer 3 Report

In the Introduction, I could not identify what is the current problem, the negative consequences of the problem and the significance to find a solution.    Methods: -only single facility used as the research setting with very small sample size which will definitely create bias and prevent from generalizability    Discussion:  very early conclusion to determine that the intervention/model is effective since the sample size is very small. 

Reviewer 4 Report

Interesting and important study in clinical practice. Patient priority care requires communication between different specialists, effective health care system. 

Introduction is described relevance of the study.

Materials and methods part clearly describe setting, Health Priorities Identification, Health priorities dentification and Implementation strategy, Implementation measurement and analysis. 

Qualitative interview with Community Living Centers-based Health Priorities Identification facilitators were explored.  Qualitative method is quite important in the manuscript, maybe authors could write as different part not merge with part 2.4. and add more information about respondents.

Result part presented comprehensive breakdown and lessons learned from implementation with two cycles and plans, study, acts and lessons learned. Qualitative data presented by themes. 

Discussion show importance of different health care professionals, project is implemented and determine outcome goal of health in practice. Study show comprehensive and high quality implementation in practice.  Qualitative study had some limitations, but it was just one part of project, besides authors qualitative interview mentioned in limitations.

Author Response

We appreciate the reviewers comments related to our manuscript and summation of our article, its findings and importance. Please see comments related to improvement needed in the Methods section below.

Interesting and important study in clinical practice. Patient priority care requires communication between different specialists, effective health care system. 

Introduction is described relevance of the study.

Materials and methods part clearly describe setting, Health Priorities Identification, Health priorities dentification and Implementation strategy, Implementation measurement and analysis. 

Qualitative interview with Community Living Centers-based Health Priorities Identification facilitators were explored.  Qualitative method is quite important in the manuscript, maybe authors could write as different part not merge with part 2.4. and add more information about respondents.

Response: Please see extensive revisions to the qualitative methods section with additional information added about the interviewer, thematic approach used and interviewees/respondents.

Result part presented comprehensive breakdown and lessons learned from implementation with two cycles and plans, study, acts and lessons learned. Qualitative data presented by themes. 

Discussion show importance of different health care professionals, project is implemented and determine outcome goal of health in practice. Study show comprehensive and high quality implementation in practice.  Qualitative study had some limitations, but it was just one part of project, besides authors qualitative interview mentioned in limitations.

Reviewer 5 Report

This is an important work and provides good guidance on implementing similar initiatives elsewhere. The lessons learned and practical considerations for implementation are well described. The conclusions are well justified within the reported results. However, some clarifications of methods and potentially acknowledgements of method limitations are missing:

It would be helpful to know if the facilitators knew that the documentation will be reviewed by the national experts (Methods) and if yes, what role it might have played (Limitations). If some of experts are also supervising the CLC. I assume that all documentation was reviewed, not a select sample (not stated explicitly).

More details on qualitative interviews in Methods would be great. Where all personnel interviewed or some. (Results report 9 trained and 5 interviews, how the choice was made?) Individual or group interviews. Who conducted interviews and if it might have biased the responses.

Page 4 line 154 “Themes explored in these interviews”. It would be great to elaborate on the methods used for the analysis. If those themes were predetermined or emerged (inductive or deductive). By whom and how the analysis was conducted. Was it semi-structures or structured interviews, using an interview guide?

Were patients interviewed? Or who was representing patient voice, for example, in terms of barriers. Can it be truly said that the process is “very appropriate for veterans” (page 6 line 193) if you did not hear from veterans or their families? On Page 9 line 277 “from the perspective of staff” clarification is rightly used.

Is acceptability and feasibility used interchangeably? In the abstract, page 1 line 26, both terms are used. In the methods acceptability is not introduced.

It would be really helpful to break down “nine facilitators completed 20 HPI encounters” into numbers for the first PDSA cycle and the second PDSA cycle. Results report rates for the first cycle in percentages and then for the second cycle improved percentages, and it is not clear how many of 20 occurred at each. And if all 9 were trained in the first cycle or additional facilitators were added in the second.

Minor comments:

Would be great to have a description of a facilitator, their expertise. The 2.1 Methods paragraph describes the unit staff, whom of them is eligible to be a facilitator? It appears under 2.3.2 but maybe can be in 2.2 since a facilitator is mentioned there first.

Page 2 line 51 “identify” please check if it should be [process that] identifies, the same with elicit and determine in the same sentence

Page 2 line 72 “SNF setting”, please check if that acronym has been introduced

Page 2 line 90, the same issue where “a facilitator explores” but under 2), 3), 4) and 5) it is a different verb form.

Page 3 line 145, interdisciplinary team (IDT) acronym is introduced, however it appears earlier in the text and might be introduced there

Page 3 line 139 “The successful implementation was measured”, most likely word “successful” is unneeded here, in the Methods, as you measure any implementation and having fidelity and feasibility might not mean “success”

Page 4 line 151 “staff perceptions of the PPC processes into their workflow”, potentially a word is missing that goes with “into”, e.g., the fit of the process into workflow?

Page 4 line 160 “patients HPI discussions” HPI discussions with patients?

Page 9 line 261 “i.e. person-centered outcomes” are inappropriately described under experience of care while they belong in measures of quality of care. Patient experience and patient satisfaction are not universally considered to be the same.

Page 9 line 283 “Successful implementation”. Since this study only reports on fidelity and feasibility, discuss “success” might be to premature or success should be defined.

Page 9 line 285 “interested becoming”, most likely “in” is missing

Page 9 line 291-294 probably is erroneous as funding is reported on page 10 lines 302-303
